# Predicting Distribution of the Asian Longhorned Beetle, *Anoplophora glabripennis* (Coleoptera: Cerambycidae) and Its Natural Enemies in China

**DOI:** 10.3390/insects13080687

**Published:** 2022-07-29

**Authors:** Quan-Cheng Zhang, Jun-Gang Wang, Yong-Hui Lei

**Affiliations:** College of Agriculture, Shihezi University, Shihezi 832003, China; zhangquancheng@stu.shzu.edu.cn (Q.-C.Z.); leiyonghui@163.com (Y.-H.L.)

**Keywords:** *Anoplophora glabripennis*, *Dastarcus helophoroides*, *Dendrocopos major*, MaxEnt, climate change, natural enemy, pest management

## Abstract

**Simple Summary:**

The Asian longhorned beetle, *Anoplophora glabripennis,* is a worldwide invasive creature that has invaded the world for more than 20 years. Although previous studies have predicted the distribution range of *A. glabripennis*, this single species distribution prediction cannot provide more potential management strategies for the control of invasive organisms. In this study, we incorporated two important natural enemies (*D. helophoroides* and *D. major*) of *A. glabripennis* into the prediction model. We found that climate change led to the northward migration of the suitable areas of *A. glabripennis* and its natural enemies. In China, only the occurrence regions of *A. glabripennis* are mainly distributed in parts of Xinjiang, Xizang, and Qinghai. In other occurrence regions of *A. glabripennis*, control models of *A. glabripennis* + *D. helophoroides*, *A. glabripennis* + *D. major*, or *A. glabripennis* + *D. helophoroides* + *D. major* were found. Fortunately, part of the areas in Xinjiang, Xizang, and Qinghai are potentially suitable for *D. helophoroides* and *D. major* under future climatic conditions. Therefore, these two natural enemies may be used in these regions to control *A. glabripennis* in the future.

**Abstract:**

The Asian longhorned beetle, *Anoplophora glabripennis*, is a forestry pest found worldwide. *A. glabripennis* causes serious harm because of the lack of natural enemies in the invaded areas. *Dastarcus helophoroides* and *Dendrocopos major* are important natural enemies of *A. glabripennis*. MaxEnt was used to simulate the distribution of *D. helophoroides* and *D. major* in China, and their suitable areas were superimposed to pinpoint which regions are potentially appropriate to release or establish natural enemy populations under current and future conditions. The results showed that, with climate change, the suitable areas of *D. helophoroides* and *D. major* migrated northward; the centroid shift of *A. glabripennis* was greater than those of *D. helophoroides* and *D. major*. From current conditions to 2090, the suitable area of *A. glabripennis*, *D. helophoroides,* and *D. major* will increase by 1.44 × 10^4^, 20.10 × 10^4^, and 31.64 × 10^4^ km^2^, respectively. Northern China (e.g., Xinjiang, Gansu, and Inner Mongolia), where *A. glabripennis* causes more serious damage, is also a potentially suitable area for *D**. helophoroides* and *D**. major*, and this provides a potential strategy for the management of *A. glabripennis*. Therefore, we suggest that natural enemies should be included in the model used for predicting suitable areas for invasive pests.

## 1. Introduction

The Asian longhorned beetle, *Anoplophora glabripennis*, has strong invasiveness and destructive power as an important forestry pest [1]. It originated in China and Korea [2], and it has a wide range of host plant species, including more than 15 genera such as *Ulmus*, *Populus*, *Salix*, and *Acer* that are continuously infected by *A**. glabripennis* [3]. Since 1996, *A**. glabripennis* has been found worldwide [4]; it has successively invaded the United States (1996) [4], Austria (2001) [5], Canada (2003) [6], France (2003), Germany (2004), Italy (2007), Belgium (2008), the Netherlands (2010), Switzerland (2011), the United Kingdom (2012), Finland (2015), and Montenegro (2015) [7]. In addition, *A**. glabripennis* has been found and has invaded for more than 20 years in Xinjiang, China [8]. *A**. glabripennis* poses significant management risks to invaded cities, farmland, forests, and ecosystems [9].

Although *A**. glabripennis* adults are relatively large, the eggs, larvae, and pupae are easily hidden in logs and boxes for transmission [10]. It is by virtue of this concealment that *A**. glabripennis* has invaded the world for more than 20 years [4]. Because *A**. glabripennis* hosts are widely distributed, there are host species of *A**. glabripennis* in almost all countries or regions [3]. *A**. glabripennis* can rely on these hosts to establish new populations, and it is very difficult to completely eradicate *A**. glabripennis* [1]. Of course, successful cases of eradication have been reported. For example, it took 11 years to completely eradicate *A**. glabripennis* in the Cornuda region of northeast Italy [11]. However, it is difficult to eradicate *A**. glabripennis* completely in all invaded areas, especially because it takes a long time to eradicate a large number of trees [1]. We began to think of a sustained, natural, low-cost way of controlling *A**. glabripennis*. According to the classical biological control theory, more valuable natural enemies can be selected from the area of origin and introduced into the invaded area to control the pest population [12,13]. The latitude range of the *A**. glabripennis* invasion area is similar to that of the distribution area at the origin, which provides feasibility for introducing natural enemies from the area of origin to control the damage of *A**. glabripennis* in the invasion area [14].

Generally, it is difficult for invasive organisms to find natural enemies in new environments [15]. The invasive organisms can reproduce, survive, and occupy ecological sites rapidly [16]. We tried to find potential predators for these *A**. glabripennis*-invaded areas. We reviewed important natural enemies of *A**. glabripennis* in the local environment. In China, *Dastarcus helophoroides* and *Dendrocopos major* are two important natural enemies of *A**. glabripennis* [14,17]. *D. helophoroides*, a parasitic beetle, has a control effect of 63–76% on mature *A**. glabripennis* larvae and pupae [18]. *D. major*, an important predatory natural enemy of *A**. glabripennis*, can prey on 22% of *A**. glabripennis* larvae in trunks [19]. *A**. glabripennis* may be controlled by releasing *D. helophoroides* and attracting *D. major* [19,20].

MaxEnt is a species distribution prediction model based on the maximum entropy theory [21]. The MaxEnt model is suitable for studies of biological invasions and responses to climate change [22] because its accuracy is better than those of other models and it is easy to use. Previous studies have predicted the distribution of *A. glabripennis* in Europe [23], North America [24], Massachusetts [25], eastern Canada [26], and worldwide [27,28]. However, these studies did not analyze natural enemy factors and did not evaluate whether there are any natural enemy resources that can naturally control *A**. glabripennis* in the invaded regions.

Currently, prediction of species distribution with the niche model is basically used for single species, especially for invasive pests. When we analyze the potential distribution areas of these species, we often ignore whether there are any natural enemy resources suitable for control in the invaded areas. These analyses cannot play a more positive role in the invasion management of pests. Therefore, we believe that the distribution of natural enemies in their natural environment needs to be considered during the analyses of invasive species, to effectively evaluate the risk of pests worldwide and positively contribute to the risk management of pests.

In this study, we used the MaxEnt model, in combination with climatic factors and topographic conditions, to predict the suitable areas of *A**. glabripennis* in China. Simultaneously, we selected important natural enemies of *A**. glabripennis*, *D. helophoroides,* and *D. major* as cases to predict and analyze the suitable areas of these two natural enemies in China, and their suitable areas were superimposed to pinpoint which regions are potentially appropriate to release or establish natural enemy populations under current and future conditions.

## 2. Materials and Methods

### 2.1. Distribution Point Collection

According to the literature, news reports, and various forestry pest quarantine websites, e.g., GBIF (*A. glabripennis* (https://doi.org/10.15468/dl.aqzjdk), *D. helophoroides* (https://doi.org/10.15468/dl.zw8ter), and *D. major* (https://doi.org/10.15468/dl.sqtgy8), accessed on 2 June 2022, recorded data of other institutions, and related literature reports), the distribution points of *A. glabripennis*, *D. helophoroides*, and *D. major* in China were recorded. The longitude and latitude were located using Google Earth (international common geographic coordinate system WGS84 (World Geodetic System 1984). The rotation minute-second was in the decimal format, and the longitude and latitude were recorded. As the values of environmental climate data in the same grid are the same, the average longitude and latitude of the data with multiple distribution points in the same grid were calculated, and the average was finally used for analysis. ENMTools.pl (https://github.com/danlwarren/ENMTools, accessed on 3 June 2022) was used to trim occurrence points so that only one observation was retained in each 30 s grid cell (corresponding to the environment variable data below), to mitigate the sampling bias of the data [29]. Finally, 31 distribution points for *A. glabripennis*, 31 distribution points for *D. helophoroides*, and 276 distribution points for *D. major* were used in modeling. Finally, the distribution of *A. glabripennis*, *D. helophoroides*, and *D. major* in China was obtained (Figure 1).

### 2.2. Climate Data Collection and Environmental Variables

The climate data in this study came from the WorldClim database (https://www.worldclim.org/, accessed on 4 June 2022), which contains 19 variables (Table 1) [30]. We selected the 1970–2000 climate data as the current climate. The latest WorldClim version 2.1 (https://worldclim.org/, accessed on 4 June 2022) was used to obtain projected future climate data. We selected the average values of four Shared Socio-economic Pathways (SSPs) under the future BCC-CSM2-MR climate model to predict the suitable area changes in 2041–2060 and 2081–2100 [31]. SSPs include SSP126 (low greenhouse gas emissions), SSP245 and SSP370 (medium greenhouse gas emissions), and SSP585 (high greenhouse gas emissions). The projected future periods are 2041–2060 and 2081–2100, which represent the middle and end of this century, respectively. In addition to bioclimatic variables, we also considered terrain and plants, including elevation, slope, aspect, gm_lc_v3, gm_ve, and veg [32]. We downloaded the elevation data from the Geospatial Data Cloud (http://www.gscloud.cn/, accessed on 4 June 2022), the GlobalMaps LandCover v3 and GlobalMaps Vegetation_v2 data from the Global Map data archives (https://globalmaps.github.io/, accessed on 4 June 2022), and the vegetation data from the Resource and Environment Science and Data Center (https://www.resdc.cn/, accessed on 4 June 2022). Then, we used the software ArcGIS version 10.4.1 (ESRI, Redlands, CA, USA, https://www.arcgis.com/, accessed on 4 June 2022) to obtain the data. The spatial resolution of all variables was 30 s, and ArcGIS version 10.4.1 was used to obtain China-wide data.

To reduce the autocorrelation between the variable data, we used ArcGIS software to load all of the variable data and performed Pearson correlation analysis on variables via multivariate and band collection statistics in the software [33]. For the factor of phase coefficient |R| > 0.8 between variables, according to the contribution rate of each environmental variable to the MaxEnt model and the replacement important value, the variables with a larger contribution rate and replacement important value are preferred to participate in MaxEnt modeling and prediction, to avoid overfitting [34]. Therefore, we selected Bio1, Bio9, Bio14, Bio20, Bio21, Bio23, and Bio25 as the environmental variables of the *A. glabripennis* distribution model (Appendix A). Bio1, Bio6, Bio8, Bio11, Bio12, Bio21, Bio23, and Bio25 were selected as the environmental variables for constructing the distribution model of *D. helophoroides* (Appendix A). We selected Bio6, Bio9, Bio11, Bio12, Bio19, Bio21, Bio22, and Bio25 as the environment variables of the *D. major* distribution model (Appendix A).

### 2.3. Optimization of Model Parameters

We used the Kuenm package (https://github.com/marlonecobos/kuenm, accessed on 4 June 2022) to optimize the regularization multiplier and feature class parameters in the R version 3.6.3 software (Vienna, Austria, https://www.r-project.org/, accessed on 4 June 2022) [35]. These two parameters are essential for building the species distribution model with the MaxEnt version 3.4.4 software (New York, NY, USA, https://biodiversityinformatics.amnh.org/open_source/maxent/, accessed on 4 June 2022). In the modeling, 75% of the data were used as the training set. A total of 1160 candidate models, with parameters reflecting all combinations of 40 regularization multiplier settings (from 0.1 to 4, the interval is 0.1), and 29 feature class combinations were evaluated.

Model selection was based on statistical significance (partial ROC), predictive ability (low omission rates), and complexity (AICc values), in that order of priority. First, candidate models were screened to retain those that were statistically significant; second, the set of models were reduced with the omission rate criterion (i.e., <5% when possible); finally, models with the lowest delta AICc values (<2) were selected among the significant and low-omission candidate models.

### 2.4. Species Distribution Model

We established the species distribution model with MaxEnt version 3.4.4 and projected it to the two future periods. The obtained model of each species had different regularization multiplier and feature class parameters. In this study, 75% of the geographic distribution data were used to train and 25% were used to validate the model. Our analysis included 10 replicates. The relative contribution of each environmental variable was evaluated using the Jackknife test [36,37].

### 2.5. Change of Suitable Area under Different Climates

In the final model, the suitability maps have values ranging from 0 to 1, indicating the probability (*p*) of occurrence of the pest. MTSPS (Maximum Test Sensitivity Plus Specifificity threshold) was used as the breakpoint [38]. Combined with the distribution data and the fitting of species-suitable areas, the criteria for classification of habitat suitability according to the existence probability were as follows: *A. glabripennis* MTSPS = 0.0639; *D. helophoroides* MTSPS = 0.0612; *D. major* MTSPS = 0.1389 [39]. Therefore, suitable *p* > 0.0639 and unsuitable *p* ≤ 0.0639 for *A. glabripennis*; suitable *p* > 0.0612 and unsuitable *p* ≤ 0.0612 for *D. helophoroides*; suitable *p* > 0.1389 and unsuitable *p* ≤ 0.1389 for *D. major*.

The simulation results used the SDMtool tool to calculate the distribution centroid and area changes in different periods of scenarios [40]. ArcGIS software was used to load the SDMtoolbox toolkit. After loading successfully, the prediction results of *A**. glabripennis*, *D. helophoroides*, and *D. major* in different periods were converted into binary grid files by ArcGIS software and the SDMtools toolkit. Then, the ‘Universal Tools’ subdirectory ‘Distribution Changes Between Binary SDMs’ tool was selected in the ‘SDMTools’ module to calculate the expansion area, stability area, and contraction area of the *A**. glabripennis*, *D. helophoroides*, and *D. major* suitability area under different periods of scenarios. The ‘Centroid Changes (Line)’ tool was used to calculate the geometric center displacement of the predicted distribution in different periods, and the overall change trend of *A**. glabripennis*, *D. helophoroides*, and *D. major* suitable areas was detected.

## 3. Results

### 3.1. Reliability Analysis of Models Established for A. glabripennis, D. helophoroides, and D. major by Using MaxEnt

Based on the AICc calculated in the Kuenm (Table 2), the best model setting for *A. glabripennis* was the feature classes (FC): linear (L), quadratic (Q), and regularization multiplier (RM) equal to 0.1; the best model setting for *D. helophoroides* was the feature classes (FC): quadratic (Q), threshold (T), and regularization multiplier (RM) equal to 1.8; the best model setting for *D. helophoroides* was the feature classes (FC): quadratic (Q), product (P), and regularization multiplier (RM) equal to 3.8. ROC curve verification results for *A. glabripennis*, *D. helophoroides*, and *D. major* showed that the 10 repeated average values of AUC were 0.918, 0.824, and 0.850, respectively; these values were higher than the random AUC value (0.5) and close to 1.0, indicating that the constructed model was reliable and could be used to predict the distribution of *A. glabripennis*, *D. helophoroides*, and *D. major* in China.

### 3.2. Potential Suitable Areas for A. glabripennis, D. helophoroides, and D. major under Current and Future Climate Conditions

In China, the occurrence regions of *A. glabripennis* are distributed in Guangdong, Guangxi, Fujian, Zhejiang, Hainan, Taiwan, Jiangxi, Guizhou, Yuanan, Sichuan, Hunan, Hubei, Jiangsu, Anhui, Henan, Hebei, Shanxi, Shaanxi, Gansu, Qinghai, Xizang, Liaoning, Jilin, Heilongjiang, Ningxia, Chongqing, Tianjing, Beijing, Shanghai, and Inner Mongolia (Figure 2). Currently, the unsuitable area and suitable area for *A. glabripennis* in China account for 94.18%, and 5.82%, respectively (Figure 2a; Appendix A). By 2050, the percentages of unsuitable area and suitable area would be 93.70% and 6.29%, respectively (Figure 2b; Appendix A). By 2090, the percentages of unsuitable area and suitable area would be 94.03% and 5.95%, respectively (Figure 2c; Appendix A).

The suitable area of *D. helophoroides* is mainly in Guangdong, Guangxi, Fujian, Zhejiang, Hainan, Taiwan, Jiangxi, Guizhou, Yuanan, Hunan, Hubei, Jiangsu, Anhui, Henan, Hebei, Shanxi, Shaanxi, Gansu, Liaoning, Jilin, Heilongjiang, Ningxia, Chongqing, Tianjing, Beijing, Shanghai, East Sichuan, and parts of Xinjiang, Qinghai, Xizang, and Inner Mongolia (Figure 3). Currently, the percentages of unsuitable area and suitable area in China are 66.55% and 33.45%, respectively (Figure 3a; Appendix A). By 2050, the percentages of unsuitable area and suitable area would be 64.50% and 35.50%, respectively (Figure 3b; Appendix A). By 2090, the percentages of unsuitable area and suitable area would be 64.45% and 35.55%, respectively (Figure 3c; Appendix A).

The suitable area of *D. major* is distributed in Hubei, Jiangsu, Anhui, Henan, Hebei, Shanxi, Shaanxi, Gansu, Liaoning, Jilin, Heilongjiang, Ningxia, Chongqing, Tianjing, Beijing, Shanghai, Sichuan, and parts of Hainan, Taiwan, Guangdong, Guangxi, Zhejiang, Fujian, Hunan, Jiangxi, Xinjiang, Qinghai, Xizang, and Inner Mongolia (Figure 4). Currently, the percentages of unsuitable area and suitable area for *D. major* in China are 64.52% and 35.48%, respectively (Figure 4a; Appendix A). By 2050, the percentages of unsuitable area and suitable area would be 61.49% and 38.51%, respectively (Figure 4b; Appendix A). By 2090, the percentages of unsuitable area and suitable area would be 61.23% and 38.77%, respectively (Figure 4c; Appendix A).

### 3.3. Spatial Transfer Characteristics of Suitable Areas for A. glabripennis, D. helophoroides, and D. major

In China, from current conditions to 2050, the expansion in suitable area for *A. glabripennis* would mainly be in northeast China (e.g., Jilin, and Heilongjiang), whereas it would contract mainly in Henan, Shaanxi, Hubei, Zhejiang, and Anhui (Figure 5a). The suitable area would expand by 9.87 × 10^4^ km^2^ and contract by 5.16 × 10^4^ km^2^ (Appendix A). From 2050 to 2090, the expansion in suitable area would mainly be in Heilong and Qinghai, whereas it would contract mainly in Liaoning, Hebei, Henan, and Hubei (Figure 5b). The suitable area would expand by 1.58 × 10^4^ km^2^ and contract by 4.80 × 10^4^ km^2^ (Appendix A). The centroid transfer trajectory from Yuncheng City (currently) to Jinzhong City (2050) and Yangfu City (2090) is shown in Figure 5c.

From current conditions to 2050, the expansion in suitable area for D. helophoroides would mainly be in Jiangsu, Jilin, and Heilongjiang, whereas it would contract mainly in Hubei and Anhui (Figure 6a). The suitable area would expand by 30.22 × 10^4^ km^2^ and contract by 10.00 × 10^4^ km^2^ (Appendix A). From 2050 to 2090, the expansion in suitable area would mainly be in Xinjiang, Shanxi, Shaanxi, Inner Mongolia, Liaoning, Jilin, and Heilongjiang, whereas it would contract mainly in Jiangsu and Guizhou (Figure 6b). The suitable area would expand by 9.38 × 10^4^ km^2^ and contract by 8.92 × 10^4^ km^2^ (Appendix A). The centroid transfer trajectory from Xiangyang City (currently) to Nanyang City (2050), and Pingdingshan City (2090) is provided in Figure 6c.

From current conditions to 2050, the expansion in suitable area for *D. major* would mainly be in Shanxi, Shaanxi, Gansu, Inner Mongolia, Liaoning, Jilin, and Heilongjiang, whereas it would contract mainly in Sichaun, Hunan, Jiangxi, Jiangsu, Guangxi, and Zhejiang (Figure 7a). The suitable area would expand by 57.86 × 10^4^ km^2^ and contract by 28.47 × 10^4^ km^2^ (Appendix A). From 2050 to 2090, the expansion in suitable area would mainly be in Xinjiang, Inner Mongolia, Jilin, and Heilongjiang, whereas it would contract mainly in Sichaun, Anhui, Hunan, Jiangxi, Jiangsu, Guangxi, and Guangdong (Figure 7b). The suitable area would expand by 21.16 × 10^4^ and contract by 18.59 × 10^4^ km^2^ (Appendix A). The centroid transfer trajectory from Shiyan City (currently) to Sanmenxia City (2050), and Jincheng City (2090) is provided in Figure 7c.

### 3.4. The Distribution Overlap Region of A. glabripennis, D. helophoroides, and D. major

Under current conditions, only the occurrence regions of *A**. glabripennis* are distributed in parts of Xinjiang, Xizang, Qinghai, Shanxi, Heilongjiang, and Inner Mongolia. In other occurrence regions of *A. glabripennis*, control modes of *A**. glabripennis* + *D. helophoroides* were found in Jilin, Xinjiang, and Inner Mongolia; control modes of *A**. glabripennis* + *D. major* were found in Yunnan, Sichuan, Qinghai, Xinjiang, and Inner Mongolia; control modes of *A**. glabripennis* + *D. helophoroides* + *D. major* were found in Liaoning, Jilin, Helongjiang, Hebei, Inner Mongolia, Hebei, Shandong, Henan, Anhui, Jiangsu, Shanxi, Shaanxi, Hubei, Hunan, Jiangxi, Sichuan, Guizhou, Yunnan, Zhejiang, Guangdong, Guangxi, Fujian, and Xinjiang (Figure 8a). From the current conditions to 2090, the area of *A. glabripennis* + *D. helophoroides* would increase in Heilongjiang, the area of *A. glabripennis* + *D. major* would increase in Jiangsu, the area of *A. glabripennis* + *D. helophoroides* + *D. major* would increase in Jilin, whereas the area would decrease in Sichuan, Hubei, Shanxi, Shaanxi, Henan, Hebei, Jiangsu, and Shandong (Figure 8a–c). The centroid shift of *A**. glabripennis* was greater than those of *D. helophoroides* and *D. major* (Figure 8d).

## 4. Discussion

MaxEnt is widely used to predict the potential distribution area of invasive organisms [37], such as *Leptinotarsa decemlineata* [41], *Popillia japonica* [42], and *Sarconesia chlorogaster* [43]. These cases provide a more positive role for the management of invasive pests. In addition, CLIMEX, GARP, GIS, and other software programs are widely used to predict species distribution [44,45]. When compared with these software programs, MaxEnt can use the AUC value of the ROC curve to judge the prediction of the model [46], correct the sample deviation in the process of collecting known distribution data [47,48], and reduce spatial biases in the GBIF database for the geographical distribution of model species [49]. In this study, the values of AUC for *A**. glabripennis*, *D. helophoroides*, and *D. major* were higher than the random AUC value (0.5) and close to 1.0, indicating that the constructed model was reliable.

Climate change is an important factor that affects species distribution and transfer [50]. The global temperature is gradually increasing, which directly leads to the northward migration of species in different climatic zones and opens up new habitats [51,52]. Among insects, *C**hilo suppressalis* [53], *Drosophila melanogaster* [54], *Nezara viridula* [55], and *Bactrocera dorsalis* [56] have been confirmed to migrate northward with climate warming. Byeon [27] and Zhou [28] also showed that the climate suitability of *A**. glabripennis* would increase in the region north of 30° N and decrease in most regions south of 30° N. Our results show that the suitable areas of *A**. glabripennis*, *D. helophoroides*, and *D. major* continue to shift northward in the current–2050 and 2050–2090 stages, and especially in the current–2050 stage, the suitable area expanded mainly in northeast China.

Generally, the cold tolerance of insects plays an important role in their survival, dispersal, distribution, and population dynamics [57]. Previous studies have shown that the supercooling point range of *A. glabripennis* of the overwintering state is −2.4 °C to −27.4 °C [58]. The developmental threshold temperature of larvae was about 12 °C, and that of 1–5 instar larvae was 10 °C; the developmental speed of young larvae was the most sensitive, followed by that of old larvae; when the temperature was higher than 30 °C, the development of all instar larvae (except the first instar larvae) was inhibited [59]. In recent years, high temperatures have frequently occurred in southern China. For example, the average daily temperature from July to August at Fujian in 2020 was 35.0°C, and the highest daily temperature was 43.4 °C [60]. Obviously, the high temperature in southern China exceeded the normal development temperature range of *A. glabripennis*. According to predictions, the coldest temperature in the north will also increase [61], which will lead to a reduction in the suitable area of *A. glabripennis* in southern China (e.g., Hunan, Henan, and Southern Hebei) and expansion of the suitable area in northern China (e.g., Liaoning, Jilin, and Heilongjiang).

*D. helophoroides* populations have been widely used in southern China, effectively suppressing the population of longhorned beetles in forests [62]. However, recent studies have shown that their parasitism efficiency in southern pine forests has decreased [63,64]. The occurrence of this phenomenon is related to extreme temperature [60]. Under natural conditions, *D. helophoroides* are often subjected to ecological factors such as extreme temperatures [65,66]. In particular, high-temperature conditions have adverse effects on the individual development and reproductive characteristics of adults [67,68,69]. In addition, low-temperature conditions determine the safe wintering of *D. helophoroides* [62]. Previous studies have shown that the supercooling point of adult *D. helophoroides* in northern China is −19.14 °C to −23.9 °C, which includes northern Gansu, Hunchun, Shandong Qinhuangdao, and other areas [70,71]. This indicates that *D. helophoroides* can adapt to the cold winter climate in northern China, and these areas can become potential suitable areas for *D. helophoroides*. The natural populations have more flexible and diverse molecular mechanisms for extreme temperatures, and they have stronger heat and cold tolerance than commercial populations [70,71,72]. However, high- and low-temperature stress has always been an important factor that affects the biological and physiological processes of *D. helophoroides* and determines population stability and pest control effect [72]. Because of global warming, the extreme temperature index of each climatic region in China shows a warming trend [73], which indicates that high-temperature weather will occur frequently in the southern region and the coldest temperature in the north will also increase. This change resulted in a decrease in the control ability of *D. helophoroides* in southern China [63,64]. In this study, a potential northward diffusion trend was observed in the suitable area of *D. helophoroides*. From the current stage to 2090, the suitable area of *D. helophoroides* in southern China (e.g., Guangdong, Fujian, Sichuan, and Jiangsu) will shrink, whereas the suitable area in northern China (e.g., Xinjiang, Shanxi, Shaanxi, Inner Mongolia, Liaoning, Jilin, and Heilongjiang) will expand.

We found no reports on the effects of climate change on the ability of *D. major* to control the beetle. However, previous studies have shown that climate has an impact on the growth and hatching of *D. major* [74]. At low temperatures, *D. major* hatches over a longer period [75]. The body weight of *D. major* showed significant seasonal changes, and it increased significantly in winter [76]. The basal body metabolic rate has been reported to be related to food composition and climate [77]. In addition, climate warming shifted the range of bird activity to higher latitudes or altitudes [78], leading to the northward migration of bird breeding range [79]. In the prediction model of this study, the suitable area of *D. major* has a trend of potential diffusion to the north. The suitable area in northern China showed an expansion trend, especially in Xinjiang, Shaanxi, Shanxi, Liaoning, Heilongjiang, and other regions.

In the centroid transfer analysis, we compared the centroid transfer range of *A**. glabripennis*, *D. helophoroides*, and *D. major*. Interestingly, the latitude range of centroid transfer of *A. glabripennis* is larger than those of *D. helophoroides* and *D. major*. In China, the latitude range of the centroid transfer trajectory of *D. helophoroides* was 32°15′ N–34° N; D. major, 32° N–36° N; *A**. glabripennis*, 35° N–38°15′ N. This indicates that the centroid shift of *A**. glabripennis* is greater than those of D. helophoroides and *D. major* under future climate conditions. Whether *D. helophoroides* and *D. major* will lead to a decline in the ability to control *A. glabripennis* needs to be studied and verified.

We reviewed ecological models of suitable habitats for pests, and almost all studies only discussed the prediction of suitable habitats for single species under climatic conditions. It is worth noting that we need to re-evaluate these models from the perspective of pest control to assess whether natural enemy resources can be used to control invasive pests, which means that more than a single-species analysis is required. In this study, we superimposed the suitable areas of *A. glabripennis* and its important natural enemies, *D. helophoroides* and *D. major*, and found that, in most regions of China, niche overlaps exist between *A. glabripennis*, *D. helophoroides,* and *D. major*, except for some regions in Xinjiang, Gansu, and Inner Mongolia.

In the *A. glabripennis* + *D. helophoroides* model, the symbiotic period of *A. glabripennis* and *D. helophoroides* can be up to 4 months [80]. *D. helophoroides* can identify and locate *A. glabripennis* in different hosts by olfactory recognition of volatile α-cubane [71], parasitize the mature larvae and pupae of *A. glabripennis*, and feed on nutrients in the host [81]. The inter-forest control effect showed that the parasitic mortality rate of *A. glabripennis* was 50–70% [82], which could basically achieve the natural control effect on *A. glabripennis*. In the *A. glabripennis* + *D. major* model, *D. major* was a key factor to control the natural population of A. glabripennis [83]; *D. major* can prey on *A. glabripennis* throughout the year [84]. The predatory functional response of the two species was Holling III [17]. The numerical response of *D. major* to the predatory larvae of *A. glabripennis* was a positive density response [17]. When the population density of *A. glabripennis* was low or medium, *D. major* exhibited a strong control effect. However, when the population density was high, the control effect of *D. major* was weak [85]. In *A. glabripennis* + *D. helophoroides* + *D. major* model, we need to further study whether the control ability of *D. helophoroides* and *D. major* against *A. glabripennis* is superimposed or antagonistic. In China, areas with *D. helophoroides* or *D. major* tend to have low levels of risk. However, in areas lacking natural enemies (e.g., Xinjiang, Gansu, and Inner Mongolia), the occurrence of *A. glabripennis* is more serious [86,87]. Fortunately, most of these areas are potential habitats for natural enemies under the future climatic conditions, so local attempts can be made to control the damage of *A. glabripennis* by introducing natural enemies.

In addition, through the literature and websites, we found that *D. helophoroides* exists in China, South Korea, Japan, and other regions [88,89,90]. *D. major* is distributed in Italy, France, the UK, Finland, the United States, Japan, the Korean Peninsula, and other countries or regions (GBIF, https://www.gbif.org/, accessed on 2 June 2022). Perhaps these two natural enemies can be used in these countries or regions to control *A. glabripennis*. However, it should be noted that, during the process of introducing natural enemies, the safety of natural enemies in the introduction area should be carefully considered to avoid secondary invasion of natural enemies [91,92].

Currently, chemical and biological control technologies are the main technical means used to effectively control the damage of *A. glabripennis* [93]. We consider that the use of MaxEnt simulation to predict the suitable area of natural enemies and *A. glabripennis* would be helpful to pinpoint which regions are potentially appropriate to release or establish natural enemy populations under current and future conditions. If there are sustainably regulated natural enemies in the place of origin or invasion, then the natural enemies can be used to reduce the adverse effects of *A. glabripennis*. If there are no natural enemies in the place of origin or invasion, then natural enemies can be introduced scientifically and safely according to their suitability in the region. Although the natural control ability of natural enemies against *A. glabripennis* is decreasing with climate change, natural enemy regulation is a primitive way of regulating harmful organisms in nature and can reduce the use of chemicals. In the future, biological and chemical control can be appropriately combined to prevent large-scale deforestation and pesticide usage. This is of great significance for establishing the sustainable regulation of *A. glabripennis*.

## 5. Conclusions

Overall, we used MaxEnt to predict the potential suitable areas of *A. glabripennis*, *D. helophoroides*, and *D. major* in China under current and future climate conditions and to analyze changes in the suitable areas and centroid transfer path. We further overlaid the suitable areas of *A. glabripennis*, *D. helophoroides*, and *D. major* and pinpointed which regions are potentially appropriate to release or establish natural enemy populations. Because of climate change, the suitable areas of *D. helophoroides* and *D. major* migrated northward; the centroid shift of *A. glabripennis* was greater than those of *D. helophoroides* and *D. major*. Northern China (e.g., Xinjiang, Gansu, and Inner Mongolia), where *A. glabripennis* causes more serious damage, is also a potential suitable area for *D. helophoroides* and *D. major*, which provides a potential strategy for the management of *A. glabripennis*. Only climate and topographical variables were considered for modeling in this study. More environmental variables such as human activity, soil type, vegetation types, and interspecies interactions should be considered in the future to improve the accuracy and precision of model prediction.

## Figures and Tables

**Figure 1 insects-13-00687-f001:**
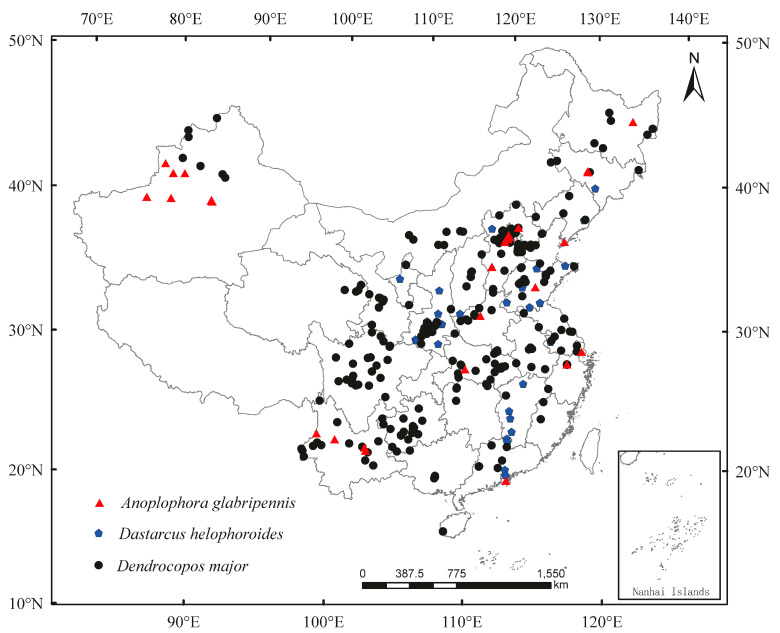
Known geographical distribution of *Anoplophora glabripennis*, *Dastarcus helophoroides*, and *Dendrocopos major* in China.

**Figure 2 insects-13-00687-f002:**
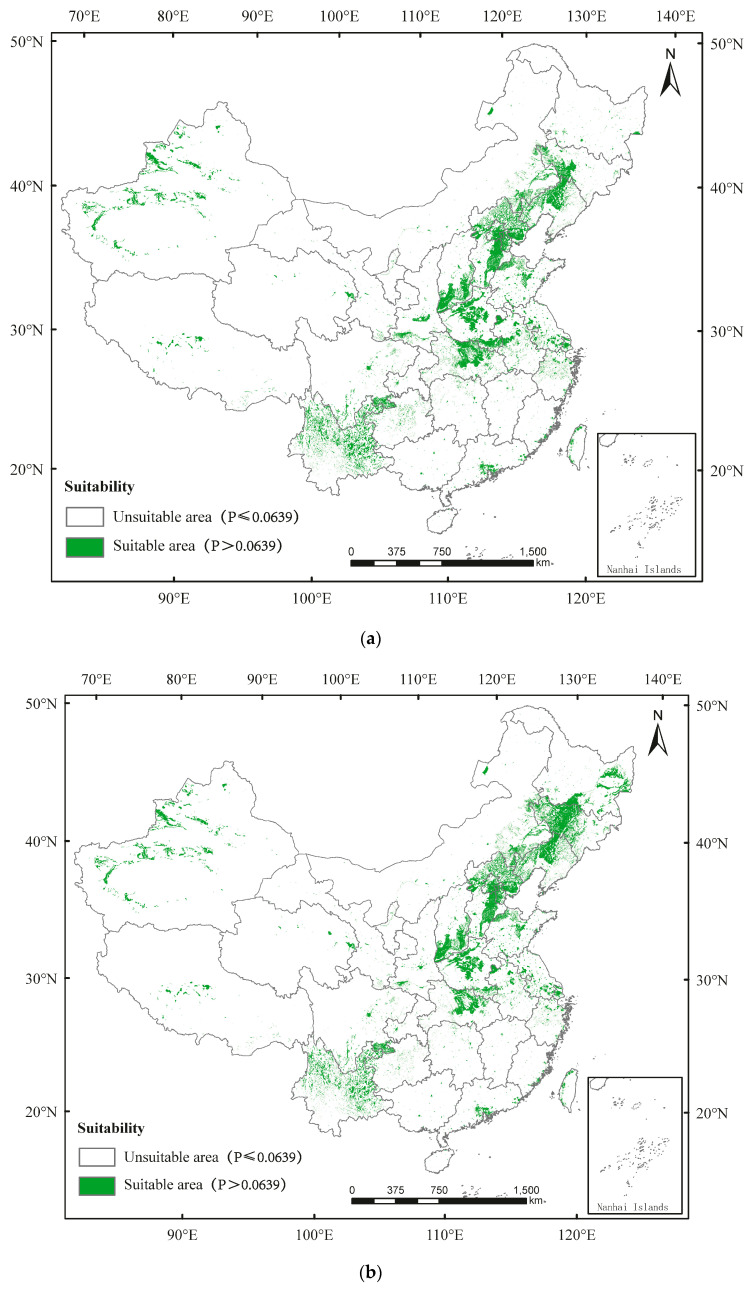
Potential distribution of *Anoplophora glabripennis* in China predicted using the MaxEnt model under (**a**) the current climate and according to climate change in (**b**) 2050 and (**c**) 2090.

**Figure 3 insects-13-00687-f003:**
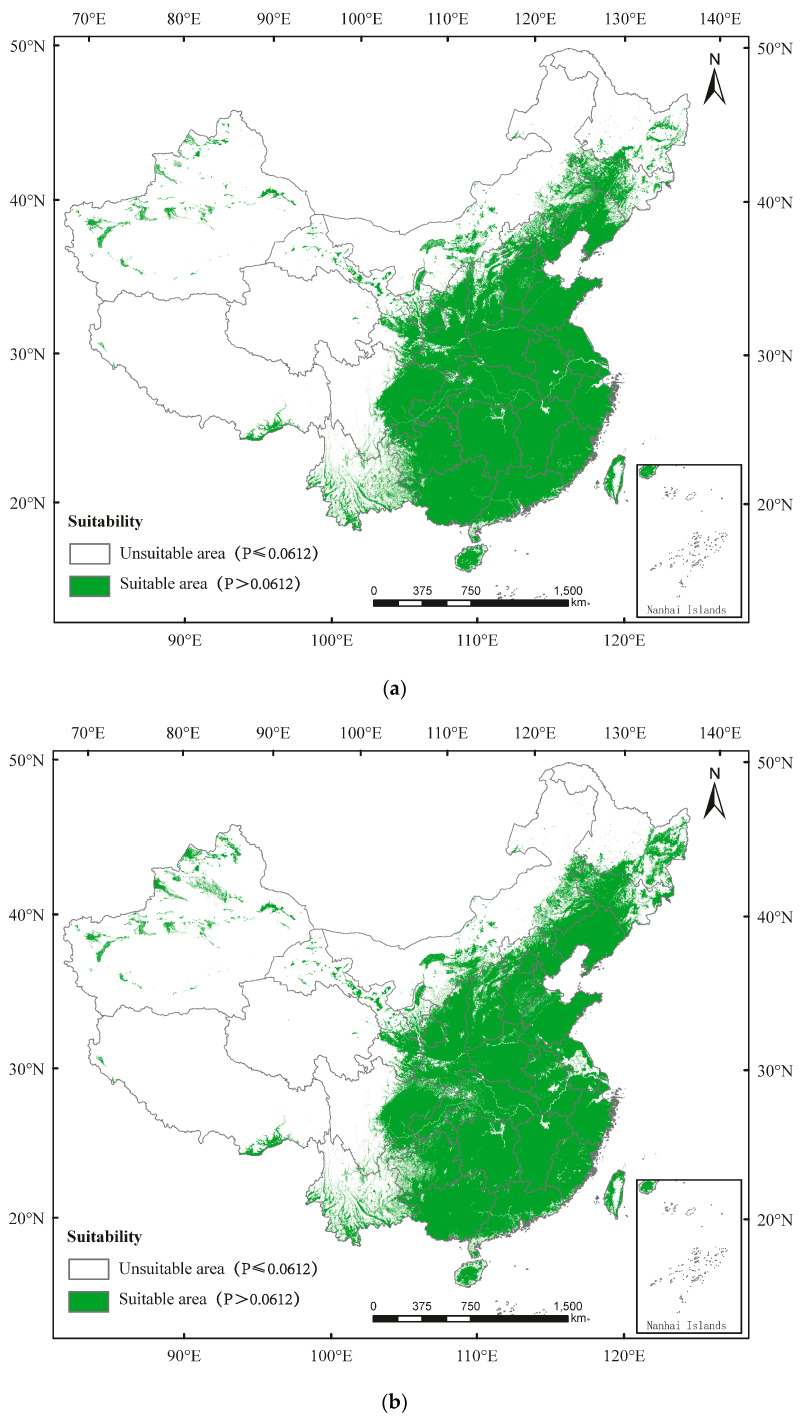
Potential distribution of *Dastarcus helophoroides* in China predicted using the MaxEnt model under (**a**) the current climate and according to climate change in (**b**) 2050 and (**c**) 2090.

**Figure 4 insects-13-00687-f004:**
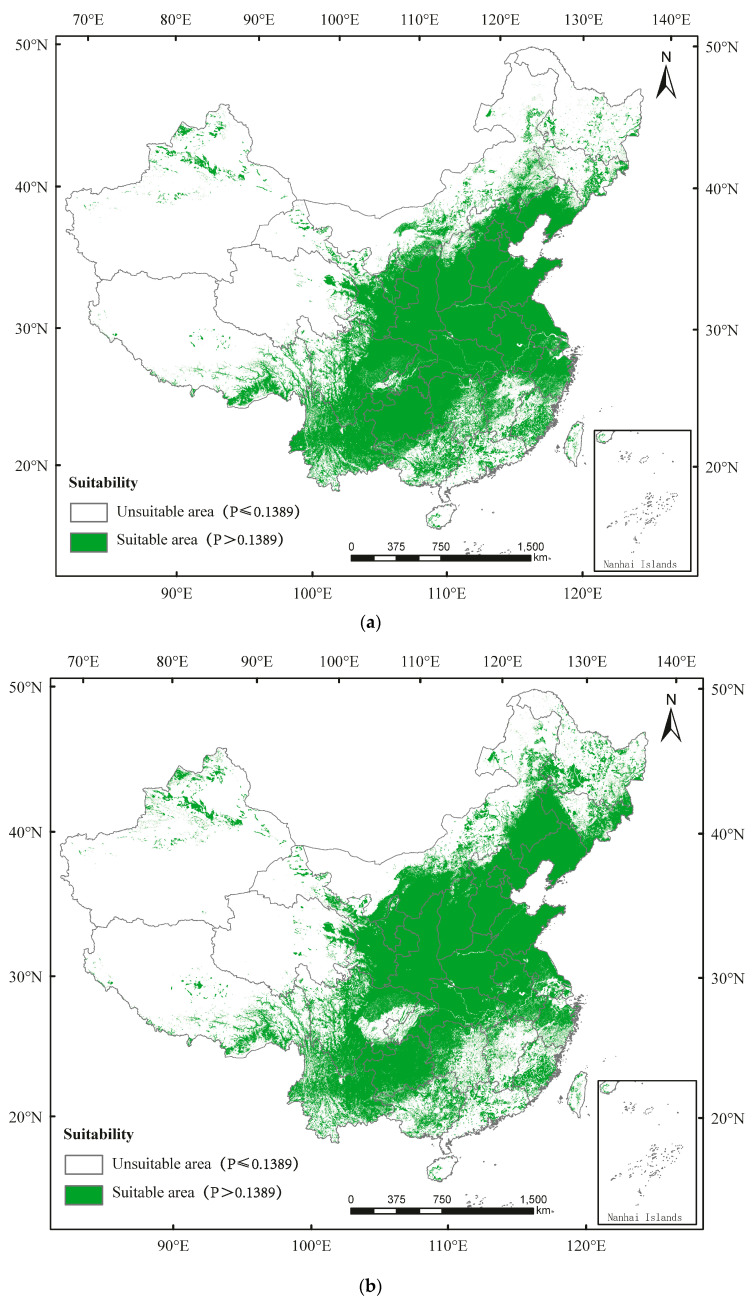
Potential distribution of *Dendrocopos major* in China predicted using the MaxEnt model under (**a**) the current climate and according to climate change in (**b**) 2050 and (**c**) 2090.

**Figure 5 insects-13-00687-f005:**
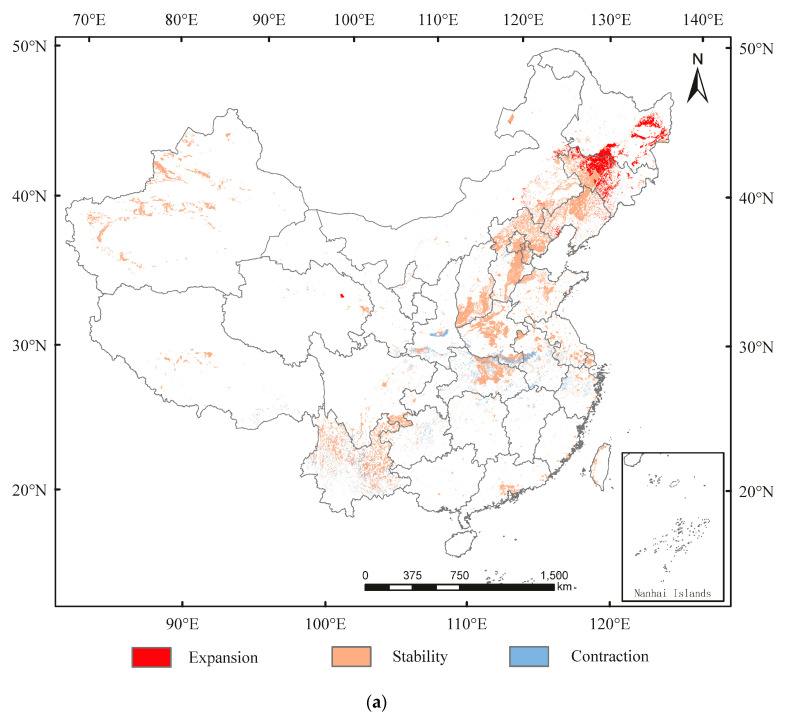
Spatial transfer characteristics of *Anoplophora glabripennis* in (**a**) current–2050 and (**b**) 2050–2090 stages and (**c**) centroid transfer trajectory in China.

**Figure 6 insects-13-00687-f006:**
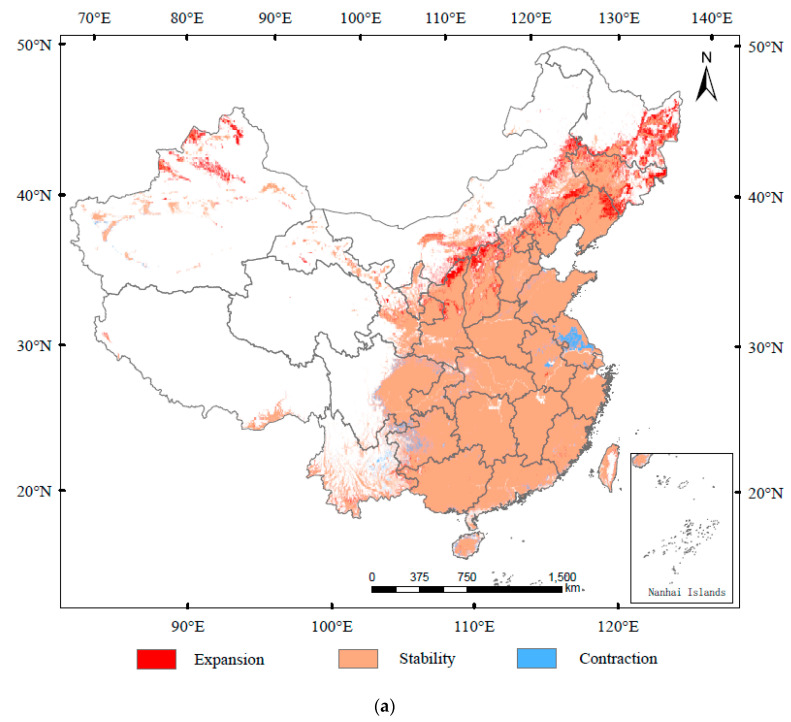
Spatial transfer characteristics of *Dastarcus helophoroides* in (**a**) current–2050 and (**b**) 2050–2090 stages and (**c**) centroid transfer trajectory in China.

**Figure 7 insects-13-00687-f007:**
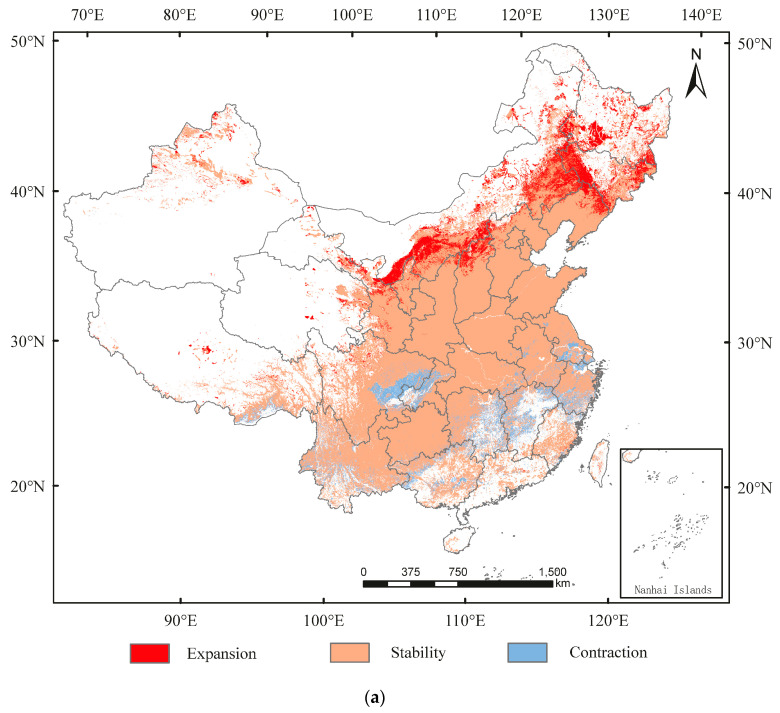
Spatial transfer characteristics of *Dendrocopos major* in (**a**) current–2050 and (**b**) 2050–2090 stages and (**c**) centroid transfer trajectory in China.

**Figure 8 insects-13-00687-f008:**
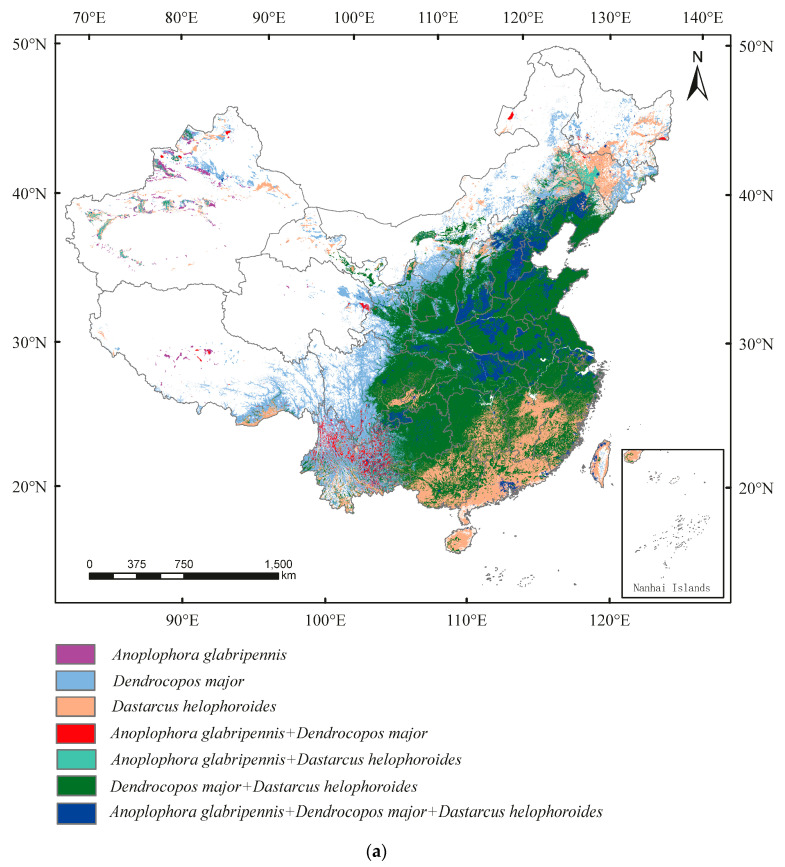
Prediction of overlapping suitable areas of *Anoplophora glabripennis*, *Dastarcus helophoroides*, and *Dendrocopos major* in (**a**) current, (**b**) 2050, and (**c**) 2090 stages and (**d**) centroid transfer trajectory in China.

**Table 1 insects-13-00687-t001:** Environment variables for MaxEnt modeling.

Environment Variable	Variable Type
Bio_1	Annual mean temperature/°C
Bio_2	Mean diurnal range/°C
Bio_3	Isothermally [(Bio2/Bio7) × 100]
Bio_4	Temperature seasonality
Bio_5	Maximum temperature of the warmest month/°C
Bio_6	Minimum temperature of the coldest month/°C
Bio_7	Temperature annual range (Bio5–Bio6)/°C
Bio_8	Mean temperature of the wettest quarter/°C
Bio_9	Mean temperature of the driest quarter/°C
Bio_10	Mean temperature of the warmest quarter/°C
Bio_11	Mean temperature of the coldest quarter/°C
Bio_12	Annual precipitation/mm
Bio_13	Precipitation of the wettest period/mm
Bio_14	Precipitation of the driest period/mm
Bio_15	Precipitation seasonality (CV)
Bio_16	Precipitation of the wettest quarter/mm
Bio_17	Precipitation of the driest quarter/mm
Bio_18	Precipitation of the warmest quarter/mm
Bio_19	Precipitation of the coldest quarter/mm
Bio_20	Aspect
Bio_21	Elevation/m
Bio_22	Slope
Bio_23	GlobalMaps°LandCover v3°
Bio_24	GlobalMaps°Vegetation v2
Bio_25	Vegetation

**Table 2 insects-13-00687-t002:** Akaike information criterion calculated in the Kuenm evaluation to select the best model setting.

Parameter	Species	FC	RM	Training AUC	Test AUC	AICc
Default	*Anoplophora glabripennis*	LQPH	1	0.9933	0.8962	237.61
	*Dastarcus_helophoroides*			0.9669	0.8192	160.69
	*Dendrocopos_major*			0.9273	0.8282	372.74
Optimized	*Anoplophora glabripennis*	LQ	0.1	0.9937	0.9179	0
	*Dastarcus_helophoroides*	QT	1.8	0.9800	0.8238	0
	*Dendrocopos_major*	QP	3.8	0.9321	0.8504	0

Abbreviations: AICc, Akaike information criterion; FC, feature classes; RM, regularization multiplier; H, hinge; L, linear; P, product; Q, quadratic; T, threshold; AUC, Area Under Curve.

## Data Availability

The data that support the findings of this study are available from the corresponding author upon reasonable request.

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
