# Peer review of "Predicting Distribution of the Asian Longhorned Beetle, Anoplophora glabripennis (Coleoptera: Cerambycidae) and Its Natural Enemies in China"

_insects, 2022, doi:10.3390/insects13080687_

Round 1

Reviewer 1 Report

Your manuscript used the overlap among ALB and its two natural enemies to assess the feasibility of biological control in China, it’s a nice idea, but the research method and analysis process seemed to be a bit simple and tediously long, which failed to prove the final conclusion your title expressed. Your manuscript would be greatly improved if more deep analyses based on the target species’ physiological characteristics were added.

Major points:

1. Throughout your manuscript, you conducted your prediction and analysis totally based on the statistical results of R packages and maxent, rather than the biological characteristics of the three target species, which may bring the risk of departure from the natural truth. I suggested to analysis the physiological needs while building your models, as well as analyzing the interaction patterns between pests and enemies in the analysis of overlapping areas.

2. You used the KUENM package to optimize your model hyperparameters, but I didn’t find the optimized results and I a bit doubt your final predictive results. Such as your simulation on ALB (Fig 2 (a)), seems to underestimate the actual distribution, as I know, the pest seriously outbreaks in northern China, such as inner Mongolia, Gansu province, and many regions your predicted results failed to cover. The AUC value of 0.801 was not very high. As I know little about the other two enemies, but I think their predictive accuracy should also be checked and improved.

3. I didn’t understand how the centroid of your predictive results are produced? Do you treat your predicted binary regions as homogenous and obtained their geometric center, or are you considered the density of potential occurrences? If it’s the first one, I think the shifts of the centroid cannot represent the shifts of the total distribution, which may also be caused by the expansion of the northern parts or the contraction of the southern parts independently. Besides, from the spatial transfer characteristics you provided (Fig8, 9, 10), I observed that the suitable areas of the three species mainly expand northward, but changed slightly on the south side. In this case, I disagree with your final conclusion. If you could provide the overlap changes of the three species (definite statistic data), it might be much clear.

Minor points:

L25: Xinjiang and China are not separate areas but subordinate, I disagree you listed them as two different regions, not only here, but the whole paper, which may bring some misunderstands. Please clarify the administrative relations between them, and revised your expression and figures.

L72: Some introduction about the biological characteristics and their interaction patterns should be added here.

L42: “Ulmus, Populus, Salix…” should be Italics, please carefully check the whole pages and corrected the other same errors (e.g. L15-160). Besides, you used the initials to refer to the pest, but used Latin names to refer to the enemies, which seems to be strange, please use the same layout.

L98-104: First, the coordinates from CABI are usually the geometric center of its mentioned administrative regions, and they didn’t represent the precise occurrence locations of the target species, I suggested not to adopt them directly. Then, please correctly cite the literature and other data source. Finally, I wonder if you have conducted the necessary data georeferencing, cleaning and thinning processes to avoid the sample bias if you have done, please added the description to this part, if you didn’t, please added these steps before modeling.

L105: The points of three different species’ occurrences should be marked in different colors and different signs to make them more distinguishable.

L153: Please give the thresholds and the reason for the binary conversion, and also the reclassify standard for the four suitable gradients you mentioned in the results.

L161: A paragraph about your optimized hyperparameters from KUENM and your model performance metrics should be given before you describe your predictive results. Besides, the other paragraphs of the result parts are too tediously long, it might be clear to give these data in histograms, and more length should be given to the comparison and summary of the suitable area changes.

L309: The first paragraph should belong to the result part.

L325: The analysis of the environmental variable contribution should combine more experimental data of species’ biological characteristics

Author Response

Dear Reviewer:

Thank you very much for your review of our manuscript that entitled “Climate Change will Weaken the Ability of Natural Enemies to Control the Asian Longhorned Beetle, Anoplophora glabripennis (Coleoptera: Cerambycidae)”.

After reviewing your comments, my co-authors and I believe that they will play an important role in improving the quality and readability of our manuscript. After careful consideration, we have responded to your comments one by one:

1. R (Reviewer): Your manuscript used the overlap among ALB and its two natural enemies to assess the feasibility of biological control in China, it’s a nice idea, but the research method and analysis process seemed to be a bit simple and tediously long, which failed to prove the final conclusion your title expressed.

1. A (Author): Thank you very much for your suggestion. In the revision, we describe the method in detail. In the results, we increase the analysis of model validation and environmental variables, and describe the changes in the suitable areas.

According to your suggestion, we changed the title of manuscript, the new title named 'Predicting distribution of the Asian Longhorned Beetle, Anoplophora glabripennis (Coleoptera: Cerambycidae) and its natural enemies in China'.

2. R: Your manuscript would be greatly improved if more deep analyses based on the target species’ physiological characteristics were added.

2. A: Thank you very much for your suggestion. Yes, we added the physiological characteristics of the target species in the revision. In the discussion, we conducted a correlation analysis between the physiological characteristics of the species and the results predicted by the model. We focused on the physiological characteristics of the species related to the model, such as high temperature resistance, supercooling point, and developmental duration.

3. R: Throughout your manuscript, you conducted your prediction and analysis totally based on the statistical results of R packages and maxent, rather than the biological characteristics of the three target species, which may bring the risk of departure from the natural truth. I suggested to analysis the physiological needs while building your models, as well as analyzing the interaction patterns between pests and enemies in the analysis of overlapping areas.

3. A: Thank you very much for your suggestion. According to your suggestion, in the new revision, we included the biological characteristics of species in the discussion of species distribution and changes in the suitable areas and analyzed them in conjunction with our results. In the overlapping region analysis, we discussed three action modes of A. glabripennis+ D. helophoroides, A. glabripennis+ D. major, and A. glabripennis + D. helophoroides + D. major, which further illustrated the positive effect of incorporating natural enemies into the model prediction.

4. R: You used the KUENM package to optimize your model hyperparameters, but I didn’t find the optimized results and I a bit doubt your final predictive results.

4. A: Thank you very much for your suggestion. In the revision, we added the optimization method and results of model parameters.

5. R: Such as your simulation on ALB (Fig 2 (a)), seems to underestimate the actual distribution, as I know, the pest seriously outbreaks in northern China, such as inner Mongolia, Gansu province, and many regions your predicted results failed to cover. The AUC value of 0.801 was not very high. As I know little about the other two enemies, but I think their predictive accuracy should also be checked and improved.

5. A: Thank you very much for your suggestion. We re-examined the species in the occurrence area, re-optimized the prediction model, and improved the AUC value. In the revision, the accuracy of the prediction model was improved.

6. R: I didn’t understand how the centroid of your predictive results are produced? Do you treat your predicted binary regions as homogenous and obtained their geometric center, or are you considered the density of potential occurrences? If it’s the first one, I think the shifts of the centroid cannot represent the shifts of the total distribution, which may also be caused by the expansion of the northern parts or the contraction of the southern parts independently.

6. A: Thank you very much for your suggestion. Centroids calculate the geometric centers of geographically appropriate areas without considering species density. In the revision, we re-described the analysis of centroid transfer, and the transfer path of centroid in different periods. As you say, the expansion of the northern region and the contraction of the southern region lead to the potential northward spread of species.

7. R: Besides, from the spatial transfer characteristics you provided (Fig8, 9, 10), I observed that the suitable areas of the three species mainly expand northward, but changed slightly on the south side. In this case, I disagree with your final conclusion. If you could provide the overlap changes of the three species (definite statistic data), it might be much clear.

7. A:Thank you for your suggestion. In the revision, our main conclusions include the trend of northward expansion of species, the dynamic changes of species suitable areas, and pinpointwhich areas can introduce natural enemies under current climate conditions. Our original conclusion is that climate change weakens the ability of natural enemies to control the ALB. This conclusion needs more investigation and verification. Therefore, we amended the original conclusion.

8. R: L25: Xinjiang and China are not separate areas but subordinate, I disagree you listed them as two different regions, not only here, but the whole paper, which may bring some misunderstands. Please clarify the administrative relations between them, and revised your expression and figures.

8. A:Thank you very much for your suggestion. We retain the prediction model of China in the revision, excluding the model of Xinjiang. This may make the expression clearer and the results more intuitive. If necessary, we can submit the model of Xinjiang as a supporting material.

9. R: L72: Some introduction about the biological characteristics and their interaction patterns should be added here.

9. A:Thank you very much for your suggestion. Yes, we emphasized the biological characteristics of these species in the revision. In the Discussion section, we analyzed the correlation between biological and physiological characteristics of species and distribution prediction in detail. We further support the results of distribution prediction from the habits, heat resistance and cold resistance of species. Through this correlation analysis, we have improved the quality and readability of manuscripts to the greatest extent.

10. R: L42: “Ulmus, Populus, Salix…” should be Italics, please carefully check the whole pages and corrected the other same errors (e.g. L15-160). Besides, you used the initials to refer to the pest, but used Latin names to refer to the enemies, which seems to be strange, please use the same layout.

10. A:Thank you very much for your suggestion. We checked the Latin namesof the species in the revision and italicized. Standardize the use of academic terms.

11. R: L98-104: First, the coordinates from CABI are usually the geometric center of its mentioned administrative regions, and they didn’t represent the precise occurrence locations of the target species, I suggested not to adopt them directly. Then, please correctly cite the literature and other data source. Finally, I wonder if you have conducted the necessary data georeferencing, cleaning and thinning processes to avoid the sample bias if you have done, please added the description to this part, if you didn’t, please added these steps before modeling.

11. A:Thank you very much for your suggestion. Yes, the distribution point data obtained from CABI has been manually checked, and points located in urban areas have been deleted. We re-describe this part of the method in the revision, adding the optimization of parameters to avoid sample deviation. At the same time, we cited the correct method of literature use and data sources.

12. R: L105: The points of three different species’ occurrences should be marked in different colors and different signs to make them more distinguishable.

12. A:Thank you very much for your suggestion. In the revision, we distinguished the distribution of three species with different colors and expressed them with different symbols.

13. R: L153: Please give the thresholds and the reason for the binary conversion, and also the reclassify standard for the four suitable gradients you mentioned in the results.

13. A:Thank you very much for your suggestion. In the revision, we give the thresholds and reasons for binary conversions and add a grading standard for the fitness zone.

14. R: L161: A paragraph about your optimized hyperparameters from KUENM and your model performance metrics should be given before you describe your predictive results. Besides, the other paragraphs of the result parts are too tediously long, it might be clear to give these data in histograms, and more length should be given to the comparison and summary of the suitable area changes.

14. A:Thank you for your suggestion. Yes, we added the method of model parameter optimization in the revision. In the Result section, we re-describethe changes in the suitable habitat of the three species. The data of the change of the suitable area are reflected in the supporting materials in the form of tables. This makes the results more intuitive.

15. R: L309: The first paragraph should belong to the result part.

15. A:Thank you for your comments. According to your suggestion, in the Result, we add model validation, selection of environment variables.  

16. R: L325: The analysis of the environmental variable contribution should combine more experimental data of species’ biological characteristics

16. A:Thank you for your suggestion. Yes, in the Discussion, we combined the biological characteristics of species (e.g.development temperature, heat tolerance, supercooling point), and supported our results with the experimental data of these biological characteristics.

We have revised the manuscript very carefully. In the new version of the manuscript we accepted your suggestions and addressed your questions. We hope that you will find our revisions and answers to be satisfactory.

On behalf of myself and my co-authors, I thank you again. Your valuable comments on the manuscript have played an important positive role in improving our manuscript.

Best regards,

Quancheng Zhang

Reviewer 2 Report

General Comment:

Overall, a clear, focused manuscript.

Methods are straightforward and easy to follow 

Results are presented and discussed clearly and concisely.

Abstract/Keywords; The abstract is including the whole necessary information

Introduction: Introduction is well-structured, informative and makes the relational for carrying out the study clear.

The methodology is fine

The results are consistent with the objective of the work

The discussion is consistent with the results

 I accept it in present form

Author Response

Dear Reviewer:

Thank you very much for your review of our manuscript that entitled “Climate Change will Weaken the Ability of Natural Enemies to Control the Asian Longhorned Beetle, Anoplophora glabripennis (Coleoptera: Cerambycidae)”.

Thank you very much for your approval of the manuscript. Thank you for taking the time to review the manuscript, and thank you for your contribution to the manuscript. 

Best regards,

Quancheng Zhang

Reviewer 3 Report

General

The manuscript addresses the interaction between an important pest and its natural enemies regarding climate change and their potential distribution. However, it needs to resolve some points to be published. Check and review the redaction by a native english reviewer, errors appear in the text, some were highlighted in yellow in the manuscript. Section 2.4 should be reviewed to standardize verb tenses. Authors use migration out of its ecological meaning, dispersal may be a better term for changes in the potential species distribution toward northern areas of China. Figures (maps) have low quality and are hard to interpret. It is suggested to increase the image resolution. Discussion section is separated from results, yet, some results related to model performance and variable importance are presented there and should be moved to the results section. Enlarge discussion by comparing your results with similar research, as presented, discussion is relatively short. Should include a final paragraph with main conclusions.

Title: correct heading. Title is misleading, comparing the potential distribution of a host and their natural enemies NE is just a small piece of information regarding the capacity to exert control of a pest. The analysis only shows changes in the distribution of the host and their NE, the potential of presence does not necessarily means that when present the NE would exert control.  

Abstract, Simple Summary

Rewrite abstract and Simple Summary according to changes in manuscript content

Introduction

It is not clear why the study was split in China and one region, Xinjiang. There is no rationale because the province is within China.  

Methodology

Because Xinjiang is a province of China it makes no sense compute distributions twice. It is recommended to assess distribution by using all records, including those of Xinjiang in a single model. 

Clearly describe the Geographic Coordinate System used, mentioning: "... international common geographic coordinate system..." is not useful.

Fig. 1 does not have a sufficient quality to observe point types and text. 

Enlarge description of methods, for example, indicate how coverage was computed, the distribution map was binarized?

 what was the binarizing threshold?

 How coverage changes were computed?

How output of different models were combined? average?

Discussion

Results an discussion are intermingled. Separate results for model fitting and variable contribution to the model.

Section 4.4 is mostly speculative. No data is presented to support that ocurrence of natural enemies NE reduce or regulate pest populations neither the opposite, that is, the lack of NE increases pest population. Thus, authors should refrain to conclude NE act as control agents in the concurrent areas of the pest and their parasitoid/predator. What the paper shows is the potential of ocurrence, given the predictor variables, other factors are required, for example, for the ALB the presence of tree hosts.

No conclusions are given. Include a final paragraph

Supplementary data

Convert pie charts to a table, so numeric values can be visualized. As presented, is difficult to extract such information.

Additional comments and suggestions are included in the revised manuscript.

Author Response

Dear Reviewer:

Thank you very much for your review of our manuscript that entitled “Climate Change will Weaken the Ability of Natural Enemies to Control the Asian Longhorned Beetle, Anoplophora glabripennis (Coleoptera: Cerambycidae)”.

After reviewing your comments, my co-authors and I believe that they will play an important role in improving the quality and readability of our manuscript. After careful consideration, we have responded to your comments one by one:

1. R (Reviewer): The manuscript addresses the interaction between an important pest and its natural enemies regarding climate change and their potential distribution. However, it needs to resolve some points to be published.

1. A (Author): Thank you very much for your affirmation and suggestion. Your suggestion improves the quality and readability of the manuscript. In the revision, we carefully responded to your suggestions. I hope you are satisfied.

2. R: Check and review the redaction by a native english reviewer, errors appear in the text, some were highlighted in yellow in the manuscript. Section 2.4 should be reviewed to standardize verb tenses.

2. A: Thank you very much for providing comments on the manuscript. I'm very sorry we bring you confusion in expression and grammar. In the revision, we corrected grammatical errors and standardized the tense of verbs.

3. R: Authors use migration out of its ecological meaning, dispersal may be a better term for changes in the potential species distribution toward northern areas of China. Figures (maps) have low quality and are hard to interpret. It is suggested to increase the image resolution.

3. A:Thank you very much for your suggestion. We improved the resolution of the image.

4. A:Discussion section is separated from results, yet, some results related to model performance and variable importance are presented there and should be moved to the results section. Enlarge discussion by comparing your results with similar research, as presented, discussion is relatively short. Should include a final paragraph with main conclusions.

4. A:Thank you very much for your suggestion. In the revision, we adjusted and distinguished the discussions and results. In the discussion, we added the biological and physiological characteristics of the species, especially focusing on the factors such as high temperature resistance, supercooling point, and growth duration of the species. We combined these factors with the model distribution. Further, we discussed in detail the mode of action of natural enemies to controlAnoplophora glabripennis when we discussed the overlap of suitable habitats. The revised discussion is more informative.

5. R: Title: correct heading. Title is misleading, comparing the potential distribution of a host and their natural enemies NE is just a small piece of information regarding the capacity to exert control of a pest. The analysis only shows changes in the distribution of the host and their NE, the potential of presence does not necessarily means that when present the NE would exert control.  

5. A:Thank you very much for your suggestion. We changed the title to 'Predicting distribution of the Asian Longhorned Beetle, Anoplophora glabripennis(Coleoptera: Cerambycidae) and its natural enemies in China'. This seems more in line with the subject.

6. R: Abstract, Simple Summary

Rewrite abstract and Simple Summary according to changes in manuscript content

6. A: Thank you very much for your suggestion. In the revision, we rewrite the Abstract and Simple Summary based on the results.

7. R: Introduction

It is not clear why the study was split in China and one region, Xinjiang. There is no rationale because the province is within China.  

7. A: Thank you very much for your suggestion. In the revision, we retain the prediction model of China and delete the prediction model of Xinjiang in the text. If necessary, the prediction model of Xinjiang can be provided as supplementary data.

8. R: Methodology

Because Xinjiang is a province of China it makes no sense compute distributions twice. It is recommended to assess distribution by using all records, including those of Xinjiang in a single model. 

8. A: Thank you very much for your suggestion. In the revision, we retain the species distribution prediction model in China. If necessary, species distribution models in Xinjiang can be submitted as supplementary data.

9. R: Clearly describe the Geographic Coordinate System used, mentioning: "... international common geographic coordinate system..." is not useful.

9. A: Thank you very much for your suggestion. The Geographic Coordinate Systemused by international common geographic coordinate systemWGS84 (World Geodetic System 1984)

10. R: Fig. 1 does not have a sufficient quality to observe point types and text. 

10. A: Thank you very much for your suggestion. In the revision, we improved the resolution of the image, and used different colors to represent the distribution of species in Figure 1.

11. R: Enlarge description of methods, for example, indicate how coverage was computed, the distribution map was binarized?

11. A: Thank you very much for your suggestion. In the revision manuscript, we re-described the method. Coverage was computedin ArcGIS. The distribution map was binarized.

12. R: what was the binarizing threshold?

12. A: Thank you very much for your suggestion. In the modeling, the criteria for classifcation of habitat suitability according to existence probability were as follows: A. glabripennisMTSPS= 0.0639; D. helophoroidesMTSPS= 0.0612; D. major MTSPS= 0.1389.

13. R: How coverage changes were computed?

13. A: Thank you very much for your suggestion. Coverage changes were computed by SDMtool (Line 189-199).

14. R: How output of different models were combined? average?

14. A: Thank you very much for your suggestion. We overlapping the models of species in current stage. Each of species models overlapped together.

15. R: Discussion

Results an discussion are intermingled. Separate results for model fitting and variable contribution to the model.

15. A: Thank you very much for your suggestion. In the revision, we adjust the results of the Discussion section to Results.

16. R: Section 4.4 is mostly speculative. No data is presented to support that ocurrence of natural enemies NE reduce or regulate pest populations neither the opposite, that is, the lack of NE increases pest population. Thus, authors should refrain to conclude NE act as control agents in the concurrent areas of the pest and their parasitoid/predator. What the paper shows is the potential of ocurrence, given the predictor variables, other factors are required, for example, for the ALB the presence of tree hosts.

16. A: Thank you very much for your suggestion. In the discussion of the revision, we added the biological and physiological characteristics of the species (e.g. high temperature tolerance, supercooling point) and combined these factors with the prediction model to further support the conclusion of the prediction model. In the discussion of the overlapping distribution area of species, we analyzed the mode of action of natural enemies on Anoplophora glabripennis, and determined the potential areas for introducing natural enemies (e.g. Xinjiang, Gansu, Inner Mongolia).

17. R: No conclusions are given. Include a final paragraph

17. A: Thank you very much for your suggestion. We added the conclusion part to the manuscript.

18. R: Supplementary data

Convert pie charts to a table, so numeric values can be visualized. As presented, is difficult to extract such information.

18. A: Thank you for your suggestion. According to your suggestion, we change the diagram of Supplementary data into a table, and all the data are intuitively reflected in the table.

19. R: Additional comments and suggestions are included in the revised manuscript.

19. A: Thank you very much for your comments and suggestion on the manuscript. We noticed some annotation in the revised manuscript. For example, the overlap of suitable areas is predicted under current climate conditions, and unifies the use of Latin. We hope you will be satisfied with the revised manuscript.

We have revised the manuscript very carefully. In the new version of the manuscript we accepted your suggestions and addressed your questions. We hope that you will find our revisions and answers to be satisfactory.

On behalf of myself and my co-authors, I thank you again. Your valuable comments on the manuscript have played an important positive role in improving our manuscript.

Best regards,

Quancheng Zhang

Round 2

Reviewer 1 Report

Dear authors:

I have directly added my suggestions below your first responses, please refer to the content of "re-review points".

“Dear Reviewer:

Thank you very much for your review of our manuscript that entitled “Climate Change will Weaken the Ability of Natural Enemies to Control the Asian Longhorned Beetle, Anoplophora glabripennis (Coleoptera: Cerambycidae)”.

After reviewing your comments, my co-authors and I believe that they will play an important role in improving the quality and readability of our manuscript. After careful consideration, we have responded to your comments one by one:

1. R (Reviewer): Your manuscript used the overlap among ALB and its two natural enemies to assess the feasibility of biological control in China, it’s a nice idea, but the research method and analysis process seemed to be a bit simple and tediously long, which failed to prove the final conclusion your title expressed.

1. A (Author): Thank you very much for your suggestion. In the revision, we describe the method in detail. In the results, we increase the analysis of model validation and environmental variables, and describe the changes in the suitable areas.

According to your suggestion, we changed the title of manuscript, the new title named 'Predicting distribution of the Asian Longhorned Beetle, Anoplophora glabripennis (Coleoptera: Cerambycidae) and its natural enemies in China'.

2. R: Your manuscript would be greatly improved if more deep analyses based on the target species’ physiological characteristics were added.

2. A: Thank you very much for your suggestion. Yes, we added the physiological characteristics of the target species in the revision. In the discussion, we conducted a correlation analysis between the physiological characteristics of the species and the results predicted by the model. We focused on the physiological characteristics of the species related to the model, such as high temperature resistance, supercooling point, and developmental duration.

3. R: Throughout your manuscript, you conducted your prediction and analysis totally based on the statistical results of R packages and maxent, rather than the biological characteristics of the three target species, which may bring the risk of departure from the natural truth. I suggested to analysis the physiological needs while building your models, as well as analyzing the interaction patterns between pests and enemies in the analysis of overlapping areas.

3. A: Thank you very much for your suggestion. According to your suggestion, in the new revision, we included the biological characteristics of species in the discussion of species distribution and changes in the suitable areas and analyzed them in conjunction with our results. In the overlapping region analysis, we discussed three action modes of A. glabripennisD. helophoroidesA. glabripennisD. major, and A. glabripennis + D. helophoroides + D. major, which further illustrated the positive effect of incorporating natural enemies into the model prediction.

[re-review points] I was glad to read your descriptions linking the biological characteristics of the target species and your predictive distributions, which make your results more convincing.

4. R: You used the KUENM package to optimize your model hyperparameters, but I didn’t find the optimized results and I a bit doubt your final predictive results.

4. A: Thank you very much for your suggestion. In the revision, we added the optimization method and results of model parameters.

5. R: Such as your simulation on ALB (Fig 2 (a)), seems to underestimate the actual distribution, as I know, the pest seriously outbreaks in northern China, such as inner Mongolia, Gansu province, and many regions your predicted results failed to cover. The AUC value of 0.801 was not very high. As I know little about the other two enemies, but I think their predictive accuracy should also be checked and improved.

5. A: Thank you very much for your suggestion. We re-examined the species in the occurrence area, re-optimized the prediction model, and improved the AUC value. In the revision, the accuracy of the prediction model was improved.

6. R: I didn’t understand how the centroid of your predictive results are produced? Do you treat your predicted binary regions as homogenous and obtained their geometric center, or are you considered the density of potential occurrences? If it’s the first one, I think the shifts of the centroid cannot represent the shifts of the total distribution, which may also be caused by the expansion of the northern parts or the contraction of the southern parts independently.

6. A: Thank you very much for your suggestion. Centroids calculate the geometric centers of geographically appropriate areas without considering species density. In the revision, we re-described the analysis of centroid transfer, and the transfer path of centroid in different periods. As you say, the expansion of the northern region and the contraction of the southern region lead to the potential northward spread of species.

7. R: Besides, from the spatial transfer characteristics you provided (Fig8, 9, 10), I observed that the suitable areas of the three species mainly expand northward, but changed slightly on the south side. In this case, I disagree with your final conclusion. If you could provide the overlap changes of the three species (definite statistic data), it might be much clear.

7. A:Thank you for your suggestion. In the revision, our main conclusions include the trend of northward expansion of species, the dynamic changes of species suitable areas, and pinpoint which areas can introduce natural enemies under current climate conditions. Our original conclusion is that climate change weakens the ability of natural enemies to control the ALB. This conclusion needs more investigation and verification. Therefore, we amended the original conclusion.

[re-review points] Your revision about the transfer of centroid was much better. But, your figure 8 overlapped the three distributions, but for which time period?  Maybe you can try to overlap the distribution of the three species for each time period and compare them, and the comparison of centroids within a same coordinate system may also help explain.

8. R: L25: Xinjiang and China are not separate areas but subordinate, I disagree you listed them as two different regions, not only here, but the whole paper, which may bring some misunderstands. Please clarify the administrative relations between them, and revised your expression and figures.

8. A:Thank you very much for your suggestion. We retain the prediction model of China in the revision, excluding the model of Xinjiang. This may make the expression clearer and the results more intuitive. If necessary, we can submit the model of Xinjiang as a supporting material.

9. R: L72: Some introduction about the biological characteristics and their interaction patterns should be added here.

9. A:Thank you very much for your suggestion. Yes, we emphasized the biological characteristics of these species in the revision. In the Discussion section, we analyzed the correlation between biological and physiological characteristics of species and distribution prediction in detail. We further support the results of distribution prediction from the habits, heat resistance and cold resistance of species. Through this correlation analysis, we have improved the quality and readability of manuscripts to the greatest extent.

10. R: L42: “Ulmus, Populus, Salix…” should be Italics, please carefully check the whole pages and corrected the other same errors (e.g. L15-160). Besides, you used the initials to refer to the pest, but used Latin names to refer to the enemies, which seems to be strange, please use the same layout.

10. A:Thank you very much for your suggestion. We checked the Latin namesof the species in the revision and italicized. Standardize the use of academic terms.

11. R: L98-104: First, the coordinates from CABI are usually the geometric center of its mentioned administrative regions, and they didn’t represent the precise occurrence locations of the target species, I suggested not to adopt them directly. Then, please correctly cite the literature and other data source. Finally, I wonder if you have conducted the necessary data georeferencing, cleaning and thinning processes to avoid the sample bias if you have done, please added the description to this part, if you didn’t, please added these steps before modeling.

11. A:Thank you very much for your suggestion. Yes, the distribution point data obtained from CABI has been manually checked, and points located in urban areas have been deleted. We re-describe this part of the method in the revision, adding the optimization of parameters to avoid sample deviation. At the same time, we cited the correct method of literature use and data sources.

[re-review points] I disagree with your method of dealing with the records from CABI, as the coordinates this website provides didn’t represent the accurate location but the center of the province, which is very inaccurate (eg. The center of Xinjiang was far away from the exact occurrence points). Besides, the citations of the data source seemed to be informal, as the GBIF website provides the DOI for users’ each downloading, and you should cite it rather than the main website address. Finally, the description of the cleaning and georeferenced process was a bit arbitrary and I suggested improving it. In my sense, you should always keep your process and description explicit and replicable especially when your method was not creative enough.

12. R: L105: The points of three different species’ occurrences should be marked in different colors and different signs to make them more distinguishable.

12. A:Thank you very much for your suggestion. In the revision, we distinguished the distribution of three species with different colors and expressed them with different symbols.

[re-review points] Your revised figure looked much clearer, please give the number of points for each species used for modeling or given in the context.

13. R: L153: Please give the thresholds and the reason for the binary conversion, and also the reclassify standard for the four suitable gradients you mentioned in the results.

13. A:Thank you very much for your suggestion. In the revision, we give the thresholds and reasons for binary conversions and add a grading standard for the fitness zone.

14. R: L161: A paragraph about your optimized hyperparameters from KUENM and your model performance metrics should be given before you describe your predictive results. Besides, the other paragraphs of the result parts are too tediously long, it might be clear to give these data in histograms, and more length should be given to the comparison and summary of the suitable area changes.

14. A:Thank you for your suggestion. Yes, we added the method of model parameter optimization in the revision. In the Result section, we re-describethe changes in the suitable habitat of the three species. The data of the change of the suitable area are reflected in the supporting materials in the form of tables. This makes the results more intuitive.

15. R: L309: The first paragraph should belong to the result part.

15. A:Thank you for your comments. According to your suggestion, in the Result, we add model validation, selection of environment variables.  

[re-review points] The revised second paragraph in the Discussion part can be considered deleted or moved to the Results part, as the Discussion was a bit long.

16. R: L325: The analysis of the environmental variable contribution should combine more experimental data of species’ biological characteristics

16. A:Thank you for your suggestion. Yes, in the Discussion, we combined the biological characteristics of species (e.g.development temperature, heat tolerance, supercooling point), and supported our results with the experimental data of these biological characteristics.

We have revised the manuscript very carefully. In the new version of the manuscript we accepted your suggestions and addressed your questions. We hope that you will find our revisions and answers to be satisfactory.

On behalf of myself and my co-authors, I thank you again. Your valuable comments on the manuscript have played an important positive role in improving our manuscript.

Best regards,

Quancheng Zhang

Author Response

Dear Reviewer:

We are very grateful for your contribution to the manuscript. We think your suggestion plays an important role in improving the quality and readability of the manuscript. We are pleased to see that you put forward minor suggestions for the manuscript in the second revision.

After careful consideration, we have responded to your comments one by one:

1. R (Reviewer): I was glad to read your descriptions linking the biological characteristics of the target species and your predictive distributions, which make your results more convincing.

1. A (Author): Thank you very much for your suggestion. I’m agree with you that linking the biological characteristics of the target species and predictive distributions, which make the results more convincing. Your suggestion does improve the quality and readability of manuscripts.

2. R: Your revision about the transfer of centroid was much better. But, your figure 8 overlapped the three distributions, but for which time period? Maybe you can try to overlap the distribution of the three species for each time period and compare them, and the comparison of centroids within a same coordinate system may also help explain.

2. A: Thank you very much for your suggestion. According to your suggestion, we show the overlapping adaptation areas at different stages ( Figure 8 ). The centroid transfer path is also compared. We think your suggestion is very good, and make the results are more clear.

3. R: I disagree with your method of dealing with the records from CABI, as the coordinates this website provides didn’t represent the accurate location but the center of the province, which is very inaccurate (eg. The center of Xinjiang was far away from the exact occurrence points). Besides, the citations of the data source seemed to be informal, as the GBIF website provides the DOI for users’ each downloading, and you should cite it rather than the main website address. Finally, the description of the cleaning and georeferenced process was a bit arbitrary and I suggested improving it. In my sense, you should always keep your process and description explicit and replicable especially when your method was not creative enough.

3. A: Thank you very much for your suggestion. The CABI data have been deleted and not used in the analysis. The distribution point data are selected. In this manuscript, the environmental data of 30 arcs and seconds are used, and the data scale is about 1 km. Since the values of environmental climate data in the same grid are the same, the average longitude and latitude of the data with multiple distribution points in the same grid are calculated, and the average is finally used for analysis.

The species distribution data in GBIF database is obtained by using R package 'rgbif'. The species distribution data obtained by R package 'rgbif' is consistent with the online data of GBIF. The R package code downloaded by GBIF is as follows :

install.packages("rgbif")

setwd("D:\\GBIFDATA")

Sys.setlocale('LC_ALL','C')

library(rgbif)

keys <- occ_search(scientificName = " Dastarcus helophoroides ")

keys$data[1,]

dim(keys$data)

SPEdata=keys$data[c(1:47)]

SPEdata

SPEdata <- SPEdata[-c(which(is.na

(SPEdata$decimalLatitude SPEdata$decimalLongitude)),is.na(SPEdata$species)),]

SPEdata

dim(SPEdata)

write.table(SPEdata, file = " Dastarcus helophoroides.xlsx",quote = F, sep = "\t",row.names = F)

DOI (Line 105-106):

Anoplophora glabripennis (https://doi.org/10.15468/dl.aqzjdk), Dastarcus helophoroides (https://doi.org/10.15468/dl.zw8ter), Dendrocopos major (https://doi.org/10.15468/dl.sqtgy8)

4. R: Your revised figure looked much clearer, please give the number of points for each species used for modeling or given in the context.

4. A: Thank you very much for your suggestion. The number of points for each species distribution points for modeling and analysis :

Anoplophora glabripennis : 31

Dastarcus helophoroides : 31

Dendrocopos major : 276

5. R: The revised second paragraph in the Discussion part can be considered deleted or moved to the Results part, as the Discussion was a bit long.

5. A: Thank you very much for your suggestion. We deleted the second paragraph of the Discussion according to your suggestion.

We have revised the manuscript very carefully. In the new version of the manuscript we accepted your suggestions and addressed your questions. We hope that you will find our revisions and answers to be satisfactory.

On behalf of myself and my co-authors, I thank you again. Your valuable comments on the manuscript have played an important positive role in improving our manuscript.

Best regards,

Quancheng Zhang